A secure cross-domain interaction scheme for blockchain-based intelligent transportation systems

Si Haiping 1
Li Weixia 1
Wang Qingyi 1
Cao Haohao 2
Bacao Fernando 3
Sun Changxia 1 sunchangxia@henau.edu.cn
1 College of Information and Management Science, Henan Agricultural University , Zhengzhou , China
2 College of Information Science and Engineering, Henan University of Technology , Zhengzhou , China
3 NOVA Information Management School (NOVA IMS), Universidade Nova de Lisboa , Lisboa , Portugal
Akleylek Sedat
Electronic publication date: 2023 Nov 15
Publication date: 2023
Volume: 9
Electronic Location ID: e1678
Received 2023 Apr 25; Accepted 2023 Oct 10
Copyright: © 2023 Si et al.
Copyright year: 2023
Copyright holder: Si et al.
License: This is an open access article distributed under the terms of the Creative Commons Attribution License, which permits unrestricted use, distribution, reproduction and adaptation in any medium and for any purpose provided that it is properly attributed. For attribution, the original author(s), title, publication source (PeerJ Computer Science) and either DOI or URL of the article must be cited.
License URL: https://creativecommons.org/licenses/by/4.0/

Keywords: Relay node, Cross-chain interaction, Intelligent transportation, Blockchain, RNCCP

Funding: Henan Province Key Science-technology Research Project 232102520006 and 232102210122 Key Research Project of Henan Provincial Higher Education Institution 23A520005 Henan Province Major Public Welfare Projects 201300210300 This work was supported by the Henan Province Key Science-technology Research Project under Grant No. 232102520006 and 232102210122, the Key Research Project of Henan Provincial Higher Education Institution under Grant No. 23A520005, and the Henan Province Major Public Welfare Projects under Grant No. 201300210300. The funders had no role in study design, data collection and analysis, decision to publish, or preparation of the manuscript.

==============================
In the intelligent transportation system (ITS), secure and efficient data communication among vehicles, road testing equipment, computing nodes, and transportation agencies is important for building a smart city-integrated transportation system. However, the traditional centralized processing approach may face threats in terms of data leakage and trust. The use of distributed, tamper-proof blockchain technology can improve the decentralized storage and security of data in the ITS network. However, the cross-trust domain devices, terminals, and transportation agencies in the heterogeneous blockchain network of the ITS still face great challenges in trusted data communication and interoperability. In this article, we propose a heterogeneous cross-chain interaction mechanism based on relay nodes and identity encryption to solve the problem of data cross-domain interaction between devices and agencies in the ITS. First, we propose the ITS cross-chain communication framework and improve the cross-chain interaction model. The relay nodes are interconnected through libP2P to form a relay node chain, which is used for cross-chain information verification and transmission. Secondly, we propose a relay node secure access scheme based on identity-based encryption to provide reliable identity authentication for relay nodes. Finally, we build a standard cross-chain communication protocol and cross-chain transaction lifecycle for this mechanism. We use Hyperledger Fabric and FISCO BCOS blockchain to design and implement this solution, and verify the feasibility of this cross-chain interaction mechanism. The experimental results show that the mechanism can achieve a stable data cross-chain read throughput of 2,000 transactions per second, which can meet the requirements of secure and efficient cross-chain communication and interaction among heterogeneous blockchains in the ITS, and has high application value.

Introduction

An intelligent transportation system (ITS) generates images of a city and its transportation by accessing data from different institutional sources (e.g., vehicle terminals, infrastructure, enterprises, government, etc.) to relieve urban traffic congestion and improve road safety. However, ITS involves a significant amount of communication and data sharing among entities such as vehicles, pedestrians, and other road users. This includes sensitive information like vehicle locations, travel trajectories, and personal data. The traditional open wireless network environment poses significant challenges to secure data transmission and data trustworthiness (Zhang & Xu, 2022). During data transmission, if the data provider is maliciously attacked or compromised by nodes, such as data tampering or leakage to evade supervision, tracking vehicle travel trajectories, or breaching personal privacy information, it can disrupt traffic order and jeopardize driving safety. In addition, ITS involves multiple data providers, including governments, traffic management departments, vehicles, communication operators, etc. How to coordinate data sharing and cooperation among all parties is a challenging issue. Therefore, the secure transmission and sharing of data in ITS have become the focus of general attention in both academic research and industrial fields.

Currently, organizations in the ITS network send their data to cloud servers for processing. Because such data are distributed in different places, the security and privacy issues of the data become particularly important (Sun et al., 2014). For example, a secure framework for sensitive data storage is proposed (Han, Han & Zhang, 2019). Han, Han & Zhang (2019) proposed a secure sharing group key management protocol (SSGK) to protect the security and privacy of data sharing in cloud storage. Khan, Parkinson & Qin (2017) proposed a fog computing scheme to solve problems such as data tampering and privacy security. However, these solutions all rely on centralized cloud service providers for user authentication and data security maintenance, which makes it difficult to achieve data sharing and identity verification across institutions in the ITS network, and the trustworthiness and security of shared data during the sharing process cannot be guaranteed. It is worth mentioning that the decentralized nature of blockchain provides a natural solution to the problem of excessive reliance on trusted third parties during the data sharing process, and achieves secure data transmission (Cheng et al., 2020; Lai et al., 2021).

Blockchain (Nakamoto, 2008) is essentially an immutable and traceable hash chain, with features such as distributed accounting, P2P transmission, and consensus mechanisms. The unique data structure of blockchain endows data records with an immutable characteristic. Once the traffic data is recorded on the blockchain, it is difficult to be tampered with and forged, which ensures the credibility of traffic data. Furthermore, blockchain technology eliminates the need for traditional centralized trust entities through the application of consensus mechanisms and cryptography. Moreover, blockchain technology, through the application of consensus mechanisms and cryptography, eliminates the need for traditional centralized trust entities. Data is stored across various nodes in the network, eliminating the vulnerability of a single central server to attacks. The verification and confirmation of traffic data no longer rely on a central server but are achieved through the collective validation of nodes in the network. This decentralized feature enhances the trustworthiness and security of traffic data. Compared with relying on the trust mechanism of intermediaries, the intelligent transportation system built by blockchain can simplify the process, and improve efficiency and security.

With the application of blockchain in ITS, the security communication, authentication, trust mechanisms, and privacy protection mechanisms in ITS have all been enhanced (Wang et al., 2020b). However, the current research has not considered the integration and sharing of traffic data resources across different blockchain systems. As various data providers in the ITS may use different blockchain platforms at the underlying level, there is no unified standard among these blockchain platforms, and a common interoperability solution is lacking. This results in poor interoperability between institution chains based on different blockchains in ITS applications, leading to an issue of data isolation. Therefore, it is essential to study a secure and reliable dynamic cross-domain data interaction mechanism for different terminals, devices, and institution chains within the ITS context.

To address the aforementioned challenges, in this article, we propose a solution for secure cross-domain data interaction in ITS, which realizes data communication and cross-chain interoperability among different facilities and organizational blockchains in the intelligent transportation network, improves the efficiency of cross-chain data sharing, and ensures the security of data in the cross-chain process. The main contributions of this article are as follows: We propose an ITS data cross-chain communication framework, which improves the original “relay chain + gateway node” cross-chain interaction model and constructs a new cross-chain interaction model. In addition, we define the standard cross-chain data communication protocol for relay nodes in this cross-chain mechanism: Relay node cross-chain protocol (RNCCP protocol), as well as cross-chain transaction lifecycle. It can be used to achieve secure data interaction between multiple heterogeneous cross-domain organizational chains in ITS.

We propose a secure access solution for relay nodes based on identity-based encryption (IBE) to provide reliable identity authentication for relay nodes.

We design and implement cross-chain interaction smart contracts and algorithms for data transmission and identity authentication between relay nodes. Based on Hyperledger Fabric and FISCO BCOS blockchains, we conduct a large number of cross-chain tests on the proposed solution using the NGSIM dataset to verify its superiority.

The remaining sections of this article are organized as follows. “Related Work” discusses related research on cross-chain communication. “Technical Background” introduces the relevant technical background of the proposed solution. “Secure Cross-Chain Interoperability Solution for Intelligent Transportation Systems” introduces the ITS secure cross-domain interaction solution. “Algorithm Implementation” provides a detailed description of the implementation details of the solution. “Experimental Evaluation” reports experimental results and theoretical analysis by constructing a cross-chain experiment in intelligent transportation scenarios, demonstrating the effectiveness of the proposed cross-chain mechanism. In “Discussion”, we compare our proposed solution with existing cross-chain approaches and analyze its limitations and challenges. Finally, “Conclusion” summarizes the article.

Related work

Today, blockchain technology has shown promising prospects and has been widely applied in various fields such as the Internet of Things (IoT), healthcare, agriculture, smart cities, and intelligent transportation systems (Wang et al., 2019; Qiao et al., 2020; Pranto et al., 2021; Ibba et al., 2017; Cocîrlea et al., 2020). In the domain of ITS research, many teams have focused on exploring how to leverage blockchain to enhance the security and data sharing in ITS. Some studies (Yuan & Wang, 2016; Sharma, Moon & Park, 2017; Balzano et al., 2022) used blockchain to create a more secure transportation system. Yuan & Wang (2016) studied a blockchain-based intelligent transportation system and designed a secure and trustworthy blockchain-based seven-layer conceptual model. Sharma, Moon & Park (2017) proposed a blockchain-based intelligent city vehicle network architecture, and constructed a new distributed vehicle transmission management system. Balzano et al. (2022) constructed a parking system in blockchain-based vehicular ad-hoc networks. In addition, some studies (Zeng et al., 2020; Li et al., 2020; Yeh, Shen & Hwang, 2022; Aftab et al., 2021) used blockchain to complete data transmission and sharing. Zeng et al. (2020) used blockchain to complete the collection, processing, and transmission of intelligent traffic light data, and proposed an information redundant cutting technique based on smart contracts in combination with edge intelligence technology, which reduced communication and time costs. Li et al. (2020) studied the integrity and privacy issues of data transmission in multiple blockchain-based interconnected vehicle network scenarios. Yeh, Shen & Hwang (2022) achieved data stable access through smart contracts and blockchain oracles, and completed traffic data sharing using 5G, P2P, and IPFS distributed file systems. Aftab et al. (2021) proposed a security and dynamic access control model for IoT-based transport systems to protect and dynamically handle data sharing and communication. These studies highlight the advantages of the combination of blockchain and intelligent transportation, bringing more development opportunities for this scenario.

However, it needs to be pointed out that despite the potential of blockchain in ITS, there are still some limitations and challenges that need to be overcome. Some of these challenges include scalability, interoperability, performance, and considerations for practical implementation. In addition, the integration and sharing of information resources are essential aspects of intelligent transportation network construction. However, the possibility of different data providers using different blockchain platforms at the underlying level, leads to a lack of interoperability among traffic data and insufficient openness and sharing of data resources, resulting in data silos. Therefore, researching cross-chain interoperability mechanisms in the field of ITS becomes a crucial task.

Cross-chain technology aims to solve the exchange and interoperability of digital assets between blockchains (Schulte et al., 2019), improve the scalability of blockchain systems, and provide secure and trusted interaction solutions for independent blockchain applications while ensuring the atomicity of cross-chain transactions (Lu et al., 2019). However, current cross-chain communication solutions primarily address the issues of asset transfer and exchange between chains. For example, Pillai et al. (2021) proposed a burn-to-claim cross-chain protocol, which uses a combination of encryption mechanisms such as digital signatures and time lock to seamlessly exchange assets between networks. Liu et al. (2021) designed a cross-chain protocol using atomic swap technology with the Vickrey auction scheme as a way to achieve efficient cross-chain asset transfer. Hei et al. (2022) constructed a cross-chain exchange system (Practical AgentChain), which allows the tokens to be mapped and traded on Practical AgentChain. In addition, some studies focused on the design of cross-chain solutions for different application scenarios. For example, Wang et al. (2020a) proposed a verification and exchange method for cross-chain transactions of heterogeneous blockchains for an electricity application, which is used to achieve secure transactions of electricity business data between different application chains. Qiao et al. (2020) proposed a cross-chain communication mechanism that simplified the communication topology of heterogeneous nodes to achieve dynamic communication between healthcare alliance chains. Shao et al. (2021) proposed an IoT blockchain cross-chain communication mechanism based on identity-based encryption to address the cross-chain communication issue in IoT environments. However, in the complex traffic network context, the bottom layers of applications of various institutions are independently developed based on different blockchain platforms. Differences in the underlying standards of blockchain limit data communication in the traffic network. The existing cross-chain mechanisms cannot meet the demand for secure cross-domain data interaction between ITS devices and institutional chains. Therefore, this article aims to design a cross-chain solution suitable for intelligent transportation networks, in order to enhance interoperability among the corresponding institutions in ITS.

Technical background

This section describes the technical background required for cross-domain communication between heterogeneous facilities and institutional chains in ITS, including cross-chain communication, libP2P, and smart contracts.

Cross-chain communication

Due to the diversity of blockchain infrastructure and the lack of unified architectural standards within the industry, research on blockchain and its interoperability solutions has become challenging (Belchior et al., 2021). In recent years, with the problem of blockchain interoperability being studied in depth, the concept of cross-blockchain communication has been considered by scholars as a necessary condition for solving blockchain interoperability. Cross-chain communication is a process that provides value to cross the barriers between chains and facilitates information transfer between one or more chains. Cross-chain communication involves two blockchains: the source and target chains. Transactions are initiated through the source chain and executed by the target chain.

Many articles have analyzed and classified cross-chain solutions. In 2016, Buterin (2016) divided cross-chain solutions into three categories, including notary mechanisms, relays, and hash locking. Based on Buterin’s research, Koens & Poll (2019) divided relays into one-way and two-way relays and evaluated these cross-chain solutions. Wang (2021) classified blockchain cross-chain research into chain-based, bridge-based, and dApp-based approaches. Qasse, Abu Talib & Nasir (2019) summarized four solutions for inter-blockchain communication, including sidechains, blockchain router, smart contracts, and industry solutions.

The cross-chain communication protocol is also an important part of achieving cross-chain interoperability. The cross-chain communication protocol refers to the process of enabling blockchain interoperability by defining a series of communication data formats, protocol specifications, and consensus protocols (Meng et al., 2022). Zamyatin et al. (2021) defined a universal cross-chain protocol as a process in which a pair of chains interact with each other to achieve synchronization and consistent states between the chains and determined the primary stages of the universal cross-chain protocol. The cross-chain communication protocol is a protocol that enables processes in the blockchain to meet validity, atomicity, and timeliness requirements.

Thus, research into the implementation of cross-chain communication technology among heterogeneous blockchains is important to the development of the cross-chain field. However, the differences in underlying standards for blockchains provide opportunities and produce many challenges to cross-chain operations.

LibP2P

LibP2P (Dias & Benet, 2016) is a modular network stack and library for building P2P networks that originated from the open-source project IPFS. The modular design of libP2P also allows it to be used as a P2P network layer for building various decentralized applications. Currently, blockchain projects such as Ethereum 2.0 (Schwarz, 2019), Polkadot (Wood, 2016), Filecoin (Psaras & Dias, 2020), BitXHub (Ye et al., 2020), chose to use the libP2P library to build the network layer of their systems. LibP2P has many excellent features, such as node discovery (i.e., the ability to discover other nodes in the P2P network), routing, support for multiple data transmission protocols, protection of transmitted data privacy, and various encryption and identity authentication mechanisms. LibP2P uses an encoding scheme called “multiaddr” to unify different protocol address formats and assigns a unique network-wide ID to the nodes in the network. By using the node ID, the issue of man-in-the-middle attacks can be effectively addressed. Due to its flexibility and scalability, libP2P has been widely used in distributed file systems and the construction of decentralized applications.

In the application scenario of blockchain, libP2P is an extremely useful tool as it enables fast data transmission. In addition, within libP2P, when two peers establish a connection, it creates a bidirectional and secure communication channel, allowing the specification of protocols used to ensure secure bidirectional communication. Experiments have demonstrated that libP2P greatly simplifies the development of P2P applications and effectively addresses most inherent issues in P2P networks (Guidi, Michienzi & Ricci, 2021).

Similarly available for building P2P networks, the BitTorrent file sharing protocol (Xia & Muppala, 2010) is mainly used for the distribution and download of large-scale files. Although both libP2P and BitTorrent can be used to build a P2P network, their main application scenarios and design goals are slightly different. LibP2P is more versatile and flexible, suitable for constructing various decentralized applications and the networking layer of blockchain projects, while BitTorrent focuses on efficient sharing and downloading of large files. It is worth noting that libP2P is committed to providing interoperability, enabling nodes with different implementations to communicate with each other. This means that libP2P can be used to build networks across different blockchain platforms or distributed applications. Therefore, in this article, libP2P is selected for interconnecting communication between relay nodes.

Smart contract

The concept of smart contracts was first proposed by Nick Szabo in 1994 and is defined as “computerized transaction protocols that execute the terms of a contract” (Szabo, 1997). However, due to limitations in computing power and application scenarios in the early days, smart contracts did not receive much attention from researchers. In 2013, Buterin (2014) introduced the Ethereum blockchain platform, which provided the Turing-complete programming language Solidity for writing smart contracts, and smart contracts were then applied to the blockchain. In blockchain technology, smart contracts can be considered a specific type of software program that is deployed and executed by a distributed ledger (Kannengießer et al., 2020). Smart contracts are crucial for facilitating, verifying, and executing transactions without the involvement of third parties in the blockchain.

Smart contracts can be classified into two types: deterministic contracts and non-deterministic contracts (Morabito, 2017). Deterministic contracts, also known as oracles, do not require external information from outside the blockchain during runtime, while non-deterministic contracts, on the contrary, rely on external information. Smart contracts can be developed and deployed on different blockchain platforms, such as Ethereum, Bitcoin, Hyperledger Fabric, and FISCO BCOS (Alharby & van Moorsel, 2017). Each platform offers smart contracts with different functionalities and programming languages. Deploying contracts on consortium blockchain platforms can provide higher performance and privacy compared to public and private blockchains.

There are also various programming languages for smart contracts, such as Solidity, Vyper, Hawk, and Serpent (Varela-Vaca & Quintero, 2021). Each language has its unique advantages and suitable scenarios. For example, Solidity is the most commonly used smart contract language on Ethereum, with extensive development tools and support. Vyper (Kaleem, Mavridou & Laszka, 2020) focuses on security and ease of analysis, while Hawk (Kosba et al., 2016) interacts with the blockchain using cryptographic primitives to protect transaction privacy. Smart contracts are critical to the operation of a blockchain because even minor errors or flaws can lead to security issues. Additionally, the privacy, performance, and scalability issues involved in smart contracts are also critical (Macrinici, Cartofeanu & Gao, 2018).

Secure cross-chain interoperability solution for intelligent transportation systems

In blockchain-based ITS, different transportation organizations, infrastructure, and road testing units use different types of blockchains according to their functional requirements. For example, the transportation department and transportation enterprises must supervise and authenticate data; thus, they use a consortium blockchain network at the underlying level. Conversely, various transportation infrastructures, such as vehicles, roads, and clouds, must share a lot of dynamic data with each other and thus use a chain-on-chain collaborative storage mode at the underlying level. In addition, users or other small data sources can use a private chain network. However, the independence of different blockchains hinders data communication and interaction between transportation organizations.

This section proposes a solution for secure cross-chain data exchange in heterogeneous ITS. To address the aforementioned situation, a novel cross-chain communication architecture composed of institution chains, facility chains, and relay node chains is constructed, as shown in Fig. 1. The various heterogeneous organization chains use the cross-chain interaction mechanism based on the relay node for data sharing and interaction. The relay nodes use the IBE mechanism to securely access the chain of relay nodes and communicate securely with other nodes. The organization chain accesses the cross-chain network through its corresponding relay node, and each organization chain performs different functional responsibilities in the ITS. With non-cross-chain interactions, users of the organization chain perform business logic processing within the blockchain they are on. If a cross-chain operation is required, the relay node forwards the cross-chain transaction request, and the other organization chains cooperate to process the operation.

Figure 1 Cross-chain communication architecture for the intelligent transport network.

Heterogeneous cross-chain interaction mechanism based on relay nodes

To achieve trusted interconnection and sharing of data among various types of blockchains across regions and organizations in the ITS, as well as the high-throughput and low-latency transaction requirements of this application scenario, this article proposes a heterogeneous cross-chain interaction model based on relay nodes to meet the needs of cross-chain interoperability in intelligent transportation scenarios. The model uses a cross-chain architecture with multiple relay nodes, where each blockchain is connected to a corresponding relay node in a one-to-one manner, and each relay node only connects to one chain. The relay nodes are interconnected through libP2P for node-to-node communication, forming a relay node chain where nodes can connect and forward requests to each other. The cross-chain transactions in the communication process are defined according to the RNCCP cross-chain data transmission protocol, and data flow between different blockchains and secure contract calling are achieved via the interaction between relay nodes.

Figure 2 shows the detailed architecture of the heterogeneous cross-chain model based on relay nodes in ITS. The architecture consists of relay nodes, relay node blockchains, cross-domain institutions, corresponding application blockchains, and smart contracts.

Figure 2 Cross-chain model architecture based on relay nodes.

The relay node chain is a blockchain cross-chain network formed by interconnecting relay nodes, and after identity authentication, nodes in the network can directly interact and communicate using libP2P.

The relay node chain includes three types of relay nodes: master node, cross-chain nodes, and light nodes. In ITS, there is only one master node in the cross-chain network, which is represented by the relay node of the transportation department chain. This node has all cross-chain transaction data and corresponding proof information and conducts identity authentication and data supervision for other relay nodes that join the cross-chain network. Nodes in the cross-chain network other than the master node can be cross-chain nodes or light nodes. The difference is that nodes participating in cross-chain interaction on both sides of the transaction are cross-chain nodes, which store the cross-chain transaction data of both parties. Other nodes that do not participate in cross-chain interaction in this transaction act as light nodes, which store only transaction data related to their own cross-chain nodes.

To meet the cross-chain communication needs of ITS, the application chains of each institution must deploy corresponding cross-chain contracts. The application chains access the cross-chain network through the relay node, use cross-chain contracts to exchange data with user contracts, and receive cross-chain requests from users, thereby achieving interchain data exchange and communication. For example, in Fig. 2, we consider transport enterprise A and transport sector B as an example. These entities deploy application chain A and application chain B, respectively, and deploy the corresponding cross-chain contracts and user contracts on the blockchain. Chains A and B are connected through the relay node chain, enabling data interaction and communication between user L of enterprise A and user Z of transport sector B.

To improve cross-chain transaction validation efficiency and cross-chain storage space utilization, this model designs different storage schemes for different types of relay nodes. First, the master node in the cross-chain network stores all cross-chain transaction information that occurs on the relay node chain, while cross-chain nodes and light nodes only must store transaction data related to themselves and save block header information for subsequent verification. Then, non-cross-chain transactions that occur only on a single blockchain do not need to be processed by the relay node and are only processed, verified, and stored by the business nodes of this blockchain. This process improves the efficiency of cross-chain transaction processing and reduces unnecessary storage costs.

Secure access and communication of IBE-based relay nodes

IBE mechanism

Identity-based encryption (IBE), which is also known as identity-based cryptography, was proposed by Shamir (1985) and consists of two mechanisms: identity-based encryption and identity-based signature. In the IBE mechanism, certificates are not necessary, and the user’s identity is directly used as the public key, simplifying the key management in the public key infrastructure (PKI). However, this idea had no suitable tools for implementation until 2001, when Boneh & Franklin (2001) used the bilinear pairings of elliptic curves to develop what is considered the first practical IBE scheme. In the IBE scheme, the trusted third party is the private key generator (PKG). Given the user’s identity, the PKG verifies the authenticity of the user’s identity, and after verification, the PKG generates the user’s private key based on the user’s identity and the system master private key. Other users in the encrypted communication only must use the public key of the other party for encryption, without the must obtain the public key through a certificate.

The IBE mechanism typically consists of four algorithms (Setup, Extract, Encrypt, Decrypt). The specific algorithms are defined as follows. We let ID={ID1,ID2,…,IDn} be the set of user identities, where ID is the user’s identity number, M is the plaintext message to be encrypted, and C is the ciphertext after encryption.

1. The algorithm Setup is used for system initialization, taking a security parameter k as input, and producing the public parameters PK publicly disclosed to everyone and the master secret key MSK known only by PKG:

(1) Setup(k)→(PK,MSK)

2. The algorithm Extract is used to extract the user’s private key from the identity information and the master secret key. It takes MSK and the user’s identity ID as input and produces the user’s private key SKID as output:

(2) Extract(MSK,ID)→SKID

3. The algorithm Encrypt is used to encrypt messages so that they can only be decrypted by a user with a specific ID. It takes the public parameters PK, the plaintext message M, and the target user’s identity ID as input, and produces the ciphertext C corresponding to the encrypted message M as output:

(3) Encrypt(PK,M,ID)→C

4. The algorithm Decrypt is used to decrypt the ciphertext. The input is the ciphertext C, the user’s private key SKID and the public parameter PK, and the output is the plaintext message M:

(4) Decrypt(C,SKID,PK)→M

The encryption and decryption process of the IBE mechanism must satisfy the consistency constraint, that is, ∀M,Encrypt(PK,M,ID)=C, and Decrypt(C,SKID,PK)=M.

Relay node secure access and communication scheme

In ITS, the corresponding relay nodes of the institutional chain must undergo identity authentication before performing cross-chain operations. In the relay node access scheme based on IBE, the unique ID of the relay node is used as the public key to replace the digital certificate issued by a third party, which simplifies the authentication process.

This solution requires initialization first, and the PKG of the transportation government department runs the Setup algorithm to generate the system parameters PK {e,P,Ppub,H1,H2} and the master key MSK of the cross-chain network. e:G1×G1→G2 is called bilinear mapping, where G1 and G2 are two q-order cyclic groups; G1 is the additive group, G2 is the multiplicative group, and they satisfy the following equations: ∀P,Q∈G1,and∀a,b∈Zq,allsatisfy e(aP,bQ)=e(P,Q)ab. Then, a hash function is selected to generate keys and authenticate identities:

(5) H1:{0,1}∗→G1H2:G2→{0,1}n

when a relay node first connects to the cross-chain network, it must be authenticated by the transportation department and request a private key from the PKG in the ITS cross-chain network. This private key is used for identity verification and secure data transmission during cross-chain transactions.

By running the Extract algorithm, the PKG maps the relay node’s identity ID∈{0,1}∗ to a point QID on an elliptic curve with order q, which is the public key QID=H1(ID), and then generates the private key SKID=MSK⋅QID for the node and sends the SKID to the relay node in a secure manner.

Once the relay node has connected to the cross-chain network, its corresponding institutional chain can directly use the relay node’s identity ID for identity verification and cross-chain data communication with other institutional chains. To ensure the security of cross-chain transactions between relay nodes, the transactions must be encrypted.

The process of encrypted communication based on IBE is as follows: when relay node A of institutional chain A sends a communication message M to relay node B of institutional chain B, the proposed method calculates QID=H1(ID) using the identity ID of relay node B. Then, the algorithm randomly selects r∈Zq∗, and the Extract algorithm is executed to generate the ciphertext C=⟨U,V⟩ corresponding to M based on the PK and QID, where U=rP, V=M⊕H2(gIDr), gID=e(PKID,Ppub). After receiving the ciphertext C, relay node B runs the Decrypt algorithm, utilizing its private key SKID and PK to decrypt the C, message plaintext M=V⊕H2(e(SKID,U)). The specific encryption communication process is shown in Fig. 3.

Figure 3 IBE-based relay node encrypted communication process.

Cross-chain communication protocols for heterogeneous chains

As an important component of cross-chain design in the ITS, the cross-chain communication protocol allows heterogeneous blockchains to access the cross-chain platform without changing the underlying structure, and perform unified cross-chain operations. This article proposes a generic interchain message transmission protocol, the RNCCP, which can perform trusted cross-chain contract interoperability and data transmission. The protocol defines a unified data structure and operation object for cross-chain transactions, defines the lifecycle of cross-chain transactions between different heterogeneous blockchains in the ITS, and provides a consistent invocation interface to the upper-level cross-chain platform.

RNCCP data structure

The RNCCP protocol is applied to relay nodes and focuses on the RNCCP data structure, which provides a uniform definition of the necessary fields of the cross-chain transaction object constructed by the relay node, as shown in Fig. 4. The key fields are explained as follows.

Figure 4 RNCCP structure constructed by relay nodes.

CrossTxNo: This field contains the unique number corresponding to the cross-chain transaction and is composed of the source chain number, the destination chain number, and the self-incrementing number corresponding to that source and destination chain stitched together. The cross-chain transaction number is generated by the relay node; thus, all blockchains participating in cross-chain transactions must maintain a ledger with all blockchain numbers in the corresponding relay node.

SrcContractInfo and DestContractInfo: These fields contain the contract name and contract version information of the corresponding cross-chain contract, respectively.

CrossChainData are byte data that are parsed into JSON cross-chain transaction data corresponding to different transaction types.

Proof: This field contains legitimacy verification information of the cross-chain transaction, which must be constructed by multiple relay nodes in collaboration. The proof contains the block number corresponding to the cross-chain transaction, the Merkle root hash and the signature of the cross-chain node for this transaction. Proof is used for transaction atomicity verification, signature verification, block header verification, etc.

The relay node parses the cross-chain message initiated by the source chain, verifies the transaction and signs it; constructs the cross-chain message as a cross-chain transaction object that conforms to the RNCCP data structure; and sends it to the relay node of the destination chain for processing.

Lifecycle of cross-chain transactions

The RNCCP protocol defines the lifecycle of cross-chain transactions, with different transaction states corresponding to different stages. Figure 5 shows the state transition diagram for the cross-chain transaction lifecycle. The StartCrossTx state indicates the beginning of a cross-chain transaction, the Executed state indicates that the cross-chain transaction has been executed, and the Completed state indicates that the cross-chain transaction has been completed.

Figure 5 Cross-chain transaction state transition diagram.

The state of a cross-chain transaction is created and transitioned according to the different cross-chain steps. During the cross-chain process, the cross-chain smart contract controls the transition of the cross-chain transaction lifecycle and implements the logic of the transition between the different states. When initiating a cross-chain transaction, the “current state” field is set in the transaction object’s data structure to indicate the transaction’s state. This state can be StartCrossTx, Executed, or Completed. Each independent cross-chain transaction begins in the null state, indicating that the transaction does not exist. The cross-chain transaction is created by the cross-chain contract on Chain A and initially assigned the StartCrossTx state. Then, the cross-chain transaction is executed on Chain B’s cross-chain contract and assigned the Executed state. Finally, the entire cross-chain transaction is completed by the cross-chain contract on Chain A and assigned the Completed state.

Cross-chain interaction process

The proposed cross-chain model primarily relies on the cross-chain contracts and relay nodes of both sides for cross-chain interaction. One or more corresponding cross-chain contracts must be deployed on the application chain. The application chain accesses the cross-chain network through the relay nodes, transfers data with user contracts through cross-chain contracts, and receives cross-chain requests from users.

There are three primary types of requests in the cross-chain interaction process: contract invocation, event listening, and relay node interaction. The contract invocation phase primarily involves the user, the user contract, and the cross-chain contract, providing the user with interfaces for cross-chain interaction. Event listening is for cross-chain contracts and relay nodes, and cross-chain nodes listen to cross-chain events of the contracts to process them accordingly. Relay node interactions are node calls between the blockchains involved in the cross-chain.

Considering the example of user A in transportation company chain A that conducts a one-way cross-chain data interaction with user B in transportation department chain B, the specific cross-chain process is shown in Fig. 6.

Figure 6 Cross-chain interaction process.

Step 1: The user of chain A creates the user contract and invokes the user contract to send the cross-chain request data.

Step 2: After receiving the cross-chain request and data from the user, the user contract calls the cross-chain contract of chain A and sends a cross-chain transaction request (CCTX request), setting the state of the cross-chain transaction to “Current-State=NULL”.

Step 3: After receiving the CCTX request, the cross-chain contract of chain A creates a unique cross-chain transaction number “CrossTxNo=[SrcChainNo:DestChainNo:Number]” by splicing the source chain number, destination chain number, and incremental code. Then, the contract generates a cross-chain transaction object and sets the state of the cross-chain transaction to “Current-State=StartCrossTx”. Finally, it sends the “startCrossTxEvent” event with the cross-chain transaction object to indicate the start of the cross-chain transaction.

Step 4: The relay node corresponding to chain A listens asynchronously for events on the blockchain.

Step 5: After listening to the “startCrossTxEvent” event and the cross-chain transaction, the relay node of chain A encapsulates the cross-chain transaction according to the defined RNCCP, and transmits the encapsulated transaction data to the relay node of chain B using libP2P.

Step 6: After receiving the cross-chain message from the node of chain A, the relay node of chain B validates the proof field of the cross-chain transaction. After verification, the relay node converts the cross-chain transaction into a transaction object that can be accepted by blockchain B according to the RNCCP. Then, the relay node calls the cross-chain contract to parse the cross-chain transaction object and perform the corresponding cross-chain operation based on the parsed cross-chain object.

Step 7: The B-chain cross-chain contract executes the cross-chain operation and records it on the blockchain. After execution, the contract sends the “ExecutedEvent” event and sets “Current-State=Executed”, indicating that the cross-chain operation has been completed.

Step 8: The relay node of chain B asynchronously listens to the “ExecutedEvent” event sent by the cross-chain contract.

Step 9: The relay node of chain B encapsulates and processes the event and the result of the cross-chain transaction and then sends the cross-chain response (CCTX response) to the cross-chain node of chain A.

Step 10: After receiving the CCTX response sent by the relay node of chain B, the relay node of chain A parses and verifies the transaction according to the RNCCP, and calls the cross-chain contract to send the transaction result.

Step 11: After receiving the cross-chain transaction result, the cross-chain contract of chain A records the transaction result on the blockchain and then calls the user contract to send the result.

Step 12: After receiving the result, the user contract of chain A returns the cross-chain result to user A.

The cross-chain transaction process is then fully executed, and user B on the blockchain can query the cross-chain result by deploying user contract B on chain B.

Algorithm implementation

In the previous section, we introduced the overall cross-chain architecture and the detailed implementation scheme for the ITS. In this section, we provide a detailed introduction to the algorithm implementation based on the above solution, primarily including the implementation algorithms related to cross-chain smart contracts, relay node communication, and relay node identity authentication.

Cross-chain interaction smart contract

As the code in the blockchain can be automatically executed according to the event trigger, the execution logic of smart contracts is important. The cross-chain interaction smart contract designed in this article serves as a bridge connecting users, blockchains, and relay node chains (see Fig. 7), and primarily implements the cross-chain interoperability logic processing between heterogeneous chains. The cross-chain contract inherits the storage contract and contains two contract reference objects (ADD and STR-OPE) and the algorithm for cross-chain interaction.

Figure 7 Cross-chain interaction structure.

1. Parent contract. The storage contract defines the related cross-chain contract events, contract objects, and necessary parameters in the cross-chain process. The names and descriptions of the corresponding cross-chain contract events and parameters are shown in Table 1. The data structure of the contract objects is shown in Fig. 4 in the previous section.

Table 1 Cross-chain transaction fields.

Name	Description	
setSrcChainNoEvent	Event emitted when the source blockchain number is set	
setDestChainNoEvent	Event emitted when the destination blockchain number is set	
createCrossTxEvent	Event emitted when the cross-chain transaction is created	
startCrossTxEvent	Event emitted when source blockchain contract sent cross-chain transaction	
executedEvent	Event emitted when destination blockchain contract executed cross-chain transaction	
completedEvent	Event emitted when source blockchain received and confirmed the result of cross-chain transaction	
Owner	The person who deployed the contract	
relayNodeList	Relay node list	
Version	Cross-chain contract version information	
CrossChainTxObjMap	The mapping list of cross-chain transaction objects	
Number	Used to calculate cross-chain transaction number	

2. Contract objects. The ADD and STR-OPE contracts are reference objects. The ADD contract sets relevant information such as the set of relay nodes and blockchain numbers. The STR-OPE contract contains algorithms for processing string-type data for cross-chain contract data processing.

3. Cross-chain interaction algorithm. In order of execution, we describe the key algorithms involved in the cross-chain interaction process in detail as follows.

startCrossChainTx(): As shown in Algorithm 1, this algorithm is called by the user contract on the blockchain. Firstly, it determines the transaction type based on the “txType” field. Then, it creates a cross-chain transaction object to initiate the cross-chain transaction. Finally, it submits the startCrossChainTxEvent on the blockchain, which is provided for listening by the relay nodes. The input of this algorithm includes relevant information about the cross-chain transaction object. This algorithm corresponds to the “StartCrossTx” state in the cross-chain transaction lifecycle. The other two states, Executed and Completed, have similar execution and completion algorithms, with the only difference being the need for ledger-related operations in the cross-chain transaction execution algorithm.

Algorithm 1 User initiated cross-chain transactions.

 Input: source address, destination address, source blockchain number, destination blockchain number, destination contract function name, cross-chain data, contract version, transaction type	
   if version.equals(contract Version) then	
     if transaction type ≠ 0 and transaction type ≠ 1 then	
        Wrong transaction type	
     else	
        if transaction type = 1 then	
         user sends contract interoperability	
         // Create cross-chain object	
         crossChainTxObj ← createCrossChainTx(source blockchain number, destination blockchain number, source address, destination address, destination contract function name, cross-chain data)	
         // Add the cross-chain object to the list of cross-chain objects	
         CrossChainTxObjMap[crossChainTxObj.crossTxNo)] ← crossChainTxObj	
         // Emit “startCrossChainTxEvent” event	
         emit startCrossChainTxEvent(“startCrossChainTxEvent”, crossChainTxObj.crossTxNo)	
       else	
         Cross-chain transaction type: transfer	
       end if	
    end if	
  end if	

sendAckedTx(). As shown in Algorithm 2, this algorithm is called by the relay node to submit the cross-chain acknowledgment transaction and parse the proof information. Firstly, the algorithm checks if the caller is a relay node, and then verifies if the contract version is compliant. Next, based on different transaction statuses, it parses data and constructs a cross-chain transaction object in accordance with the RNCCP structure. Finally, the algorithm sends the sendAckedEvent on the blockchain.

Algorithm 2 Relay node submits cross-chain confirmation transaction.

 Input: crossTxNo, txResult, contract version, proof	
    // Verify if the algorithm is called by the relay node	
   if checkRelayNode() = false then	
     This function is restricted to the relay node	
   else	
     if version.equals(contract version) then	
        // Obtain cross-chain object from the list: CrossChainTxObjMap	
        crossTxObj ← CrossChainTxObjMap[crossTxNo]	
        if crossTxObj.TxResult equals to INIT then	
         if the relay node belongs to the source chain then	
          if crossTxObj.TxType equals to transfer then	
            Parsing cross-chain object and payload based on transfer type	
          else	
            // crossTxObj.TxType equals contract interoperability	
            Parsing cross-chain object and payload based on contract interoperability type	
          end if	
           crossTxObj.TxResult ← txResult	
           Parsing transaction hash and block number where the transaction is located from proof	
           crossTxObj.proof ← proof	
           CrossChainTxObjMap[crossTxNo] ← crossTxObj	
           emit sendAckedEvent(“sendAckedEvent”,crossTxObj.CrossTxNo,crossTxObj.TxType)	
         end if	
        end if	
    end if	
end if	

Relay nodes

As an important bridge that connects different regional blockchains in ITS, the cross-chain network composed of relay nodes must have the following functions: (1) secure data transmission between nodes; and (2) trusted node identity authentication. To achieve secure and interactive communication between nodes in the cross-chain network, this study uses libP2P to build a peer-to-peer communication network between nodes.

The relay node module includes communication between nodes and communication between nodes and blockchains. Communication between nodes is based on the libP2P framework, while communication between nodes and blockchains is implemented using event listening mechanisms and the blockchain SDK. The primary algorithms for node-to-node communication and node-to-blockchain communication are given below.

nodesCommunication(): As shown in Algorithm 3, this algorithm creates p2p nodes for two relay nodes A and B. We assume that node A is the server and responsible for listening, while node B is the client and responsible for connecting to the server and establishing a communication channel with the server. Node A opens a listening port, and node B connects to node A using the encoding address. Then, the two nodes can communicate in encrypted channels with multiple concurrent streams. The inputs for this algorithm are the listening port and address of node A and the host object in the P2P network. The algorithm then determines whether node A is the server or the client based on whether the node address is empty or not. If the address is empty, then it is the server, and the port listening on the server side is opened. When a client node connects, according to the protocol specified by the PID, it begins the read-and-write flow. If the address is not empty, it means that the node is a client. After parsing the address and reading the multiaddress of the server, the multiaddress of the node is added to the peerstore. Then, the communication stream is opened between the two nodes, and a non-blocking read-and-write byte stream and a read-and-write process are created to start communication.

Algorithm 3 Inter-node communication.

Input: listening port (source port), server-side node address (dest), object in p2p network (host)	
 if dest = “ ” then	
      // When a client node connects to the server, the read/write stream is started using the protocol specified by the PID, and the “streamHandler” is the non-blocking read/write buffer stream created.	
      host.SetStreamHandler(PID, streamHandler)	
      for each la in range h.Network().ListenAddresses() do	
       p ← la.valueForProtocol(multiaddr.P_TCP)	
       port ← p	
      end for	
    else	
      madder ← multiaddr.NewMultiaddr(dest)	
      // Extract peer ID from multiaddr.	
      info ← peer.AddrInfoFromP2PAddr(madder)	
      // Add the address of the server node to the peerstore.	
      host.PeerStore().AddAddrs(info.ID, info.Address, peerstore.PermanentAddrTTL)	
     // Establishing a stream with a server-side node.	
      s ← h.NewStream(context.Background (), info.ID, PID)	
     // Create a non-blocking read/write buffer stream.	
     rw ← buffer.NewReaderWriter(buffer.NewReader(s), buffer.NewWriter(s))	
     Execute writeData(rw) // Create write threads.	
     Execute readData(rw) // Create read threads.	
 end if	

contractEventListener(). As shown in Algorithm 4, this algorithm acts as a bridge for intercommunication between relay nodes and the blockchain. It utilizes an event listening mechanism to monitor events occurring in the contract. It calls back the client with the relevant cross-chain events that happened on the blockchain. Depending on different cross-chain events, it processes them accordingly. For example, when the startCrossChainTxEvent is detected, the source chain’s relay node forwards the cross-chain transaction information to the destination chain’s relay node. Similarly, when the “executedEvent” is detected, it indicates that the cross-chain operation has been executed in the destination chain’s contract, and the destination chain’s relay node forwards the relevant information back to the source chain’s relay node.

Algorithm 4 Contract event listening.

 Input: start block, end block	
    Set relevant parameters, including start block, end block, contract address, contract event, etc.	
    Construct a callback function to retrieve the observed cross-chain events: logs	
   if logs are not empty then	
     for each log in logs do	
         list ← decode event log content	
         crossTxNo ← list[1]	
         crossTxType ← list[2]	
         blockNumber ← log.getBlockNumber()	
         txHash ← log.getTransactionHash()	
         eventName ← list[0]	
         // Execute different functions based on the eventName	
        if eventName.equals(“startCrossChainTxEvent”) then	
          Execute startTx()	
        else	
             if eventName.equals(“executedEvent”) then	
                Execute execute()	
                 eventName.equals(“sendAckedEvent”)	
            else	
                Execute sendAcked()	
            end if	
        end if	
    end for	
  end if	

IBE-based authentication of relay nodes

When a relay node applies to join the network, it must communicate with the master node. After it is authenticated by the master node, it can join the cross-chain network. Once the relay node has been authenticated, its multiaddress will be added to the peerstore, and the address will be broadcast to other nodes in the network. Each node in the cross-chain network maintains a ledger with the multiple addresses of all nodes that are used for subsequent communication between the nodes.

RelayNodeAuthentication(): As shown in Algorithm 5, this algorithm is used for identity authentication of the relay node when joining the network. The algorithm is based on the IBE mechanism, which generates a public-private key pair for the node at startup and is used to encrypt and decrypt the node’s identity. Identity is authenticated by the master node corresponding to the transport department chain in the ITS. If the identity matches, the authentication is passed, and subsequent cross-chain data interaction can be performed.

Algorithm 5 Node authentication based on IBE.

 Input: relay node identity (id), system master key (master_key), system parameters (parameters)	
 Output: Whether the relay node has passed identity authentication (authenticated), session key used for subsequent communication (session_key)	
    // Generate a random number as the private key of the relay node.	
    private_key ← generate_private_key()	
    // Generate the relay node's public key	
    public_key ← derive_public_key(private_key)	
    // Use IBE to encrypt the identity of the relay node.	
    encrypted_id ← ibe_encrypt(id, public_key, parameters)	
    // The relay node sends the encrypted identity and its public key to the main node of the cross-chain network.	
    message ← {encrypted_id, public_key}	
    // The master node uses IBE to decrypt the encrypted identity.	
    decrypted_id ← ibe_decrypt(encrypted_id, private_key, parameters)	
    // If the decrypted identity matches the public key sent by the relay node, the authentication is successful.	
   if decrypted_id = public_key then	
      authenticated ← true	
      // Generate a session key for subsequent communication.	
      session_key ← generate_session_key()	
   else	
      authenticated ← false	
   end if	

Experimental Evaluation

In this section, we first introduce the experimental environment used in this article. Then, we simulate the proposed cross-chain mechanism on Hyperledger Fabric and FISCO BCOS blockchain and analyze the performance of the mechanism through experimental tests.

Setup

In order to test the feasibility and performance of the proposed solution in this article, we use Hyperledger Fabric and FISCO BCOS to build a network environment for the heterogeneous cross-chain interaction architecture based on relay nodes. The cross-chain network consisted of 19 nodes. Specifically, the HyperLedger Fabric blockchain network was set up with 11 nodes, comprising three orderer nodes and four organizations, with each organization having two peer nodes. The Raft consensus algorithm was utilized for this blockchain network. The FISCO BCOS network was deployed with three groups, totaling eight nodes, and the Raft consensus algorithm was also chosen as the consensus mechanism for this blockchain. The experimental environment is Intel Xeon E5 CPU, 64 GB of RAM, 35 MB of L3 cache, and the Ubuntu 18.04 operating system.

We wrote and deployed smart contracts for cross-chain communication and used Java and Go languages to write corresponding relay node execution algorithms for two institutional chains. Then, we utilized the NGSIM dataset to simulate a scenario of cross-chain interaction between two heterogeneous institutional chains in ITS, and Fig. 8 shows the results of the two traffic data reads across the chains. Finally, we employed Hyperledger Caliper and JMeter tools to script performance evaluation tests for the cross-chain smart contract and the overall cross-chain process.

Figure 8 Results of cross-chain reading of traffic data.

The NGSIM dataset includes four different scenarios from six cities in the United States: US-101, I-80, Lankershim, and Peachtree. US-101 and I-80 record the trajectories of vehicles on highways, while Lankershim and Peachtree record the trajectories of vehicles on city roads. The data fields in this dataset are explained in Table 2. For testing, we randomly selected data from this dataset to simulate 10,000 concurrent traffic services.

Table 2 The data fields of the NGSIM dataset.

Name	Description	
Vehicle_ID	Vehicle identification number	
Frame_ID	The frame number of the data at a certain moment	
Total_Frames	Total number of frames in which the vehicle appears in this dataset	
Global_Time	Timestamp (ms)	
Local_X	X value in the local coordinate system, measured in feet	
Local_Y	Y value in the local coordinate system, measured in feet	
Global_X	X value in the standard geographic coordinate system	
Global_Y	Y value in the standard geographic coordinate system	
v_length	Vehicle length, measured in feet	
v_Width	Vehicle width, measured in feet	
v_Class	Vehicle type: 1-Motorcycle, 2-Car, 3-Truck	
v_Vel	Instantaneous vehicle velocity, measured in feet per second	
Lane_ID	Current lane position of the vehicle	
Location	Street name or highway name	

Complexity analysis

Before presenting the numerical evaluation results, we analyze the complexity of the five algorithms in the proposed scheme. Algorithm 1 submits an event on the blockchain after creating a cross-chain transaction object. The time complexity of this algorithm can be represented as O(1). Algorithm 2 is used by relay nodes to submit cross-chain transaction information, construct transaction objects compliant with RNCCP structure, and parse Proof information. The time complexity of these operations is also O(1); therefore, the time complexity of this algorithm can be represented as O(1). Algorithm 3 is used for communication between relay nodes, assuming that the number of relay nodes is m. Nodes connect by retrieving “multiaddr” encoded addresses. Once connected, nodes can engage in concurrent communication, and connected relay nodes can communicate directly within a fixed time, avoiding unnecessary time costs for reconnection. The time complexity of establishing connections is O(m), and the time complexity of concurrent communication is O(1). Thus, the total time complexity of this algorithm is O(m). Algorithm 4 achieves communication between relay nodes and the blockchain through event listening. Assuming that the total number of cross-chain events on a single blockchain is N, and the number of cross-chain events from the start block to the end block within the specified time is n (n<<N). This algorithm iterates through n cross-chain events and executes different cross-chain event handling methods. The time complexity of executing event handling methods is O(1), so the time complexity of Algorithm 4 is O(n). Algorithm 5 mainly involves identity encryption and decryption, as well as the transmission of ciphertext between relay nodes and main nodes. Since relay nodes can only establish connections with main nodes after identity authentication, the time complexity of this algorithm is O(m).

In summary, based on Fig. 6, the cross-chain interaction process, for a blockchain that joins this cross-chain network for the first time, incurs the longest overall cross-chain time consumption, with a time complexity of O(2m+2n).

Performance evaluation

We used widely recognized metrics in the field of cross-chain research, namely execution time and gas consumption, to evaluate the performance of our approach. The experiments in this study were divided into three groups. The first group of experiments tested the overall performance of our proposed cross-chain solution, and this group of experiments ignored the impact of transaction initiation time. The second group of experiments tested the performance of the cross-chain contract algorithm, evaluating the smart contract algorithms’ performance under different levels of concurrent traffic services. The third group of experiments tested the gas consumption required for executing cross-chain operations. This was done to evaluate the computing and storage resources consumed by cross-chain transactions. This experiment was conducted on the Ethereum platform.

To make the experimental presentation more clear, we give a detailed description of the relevant experimental parameters in Table 3.

Table 3 Experimental parameter definitions.

Name	Description	
Cross-chain traffic services	A service represents a cross-chain request, where the data field of this cross-chain request corresponds to a traffic data object from the NGSIM dataset.	
Number of parallel transactions	The number of cross-chain transactions executed concurrently in the network environment.	
Transaction rounds	Due to a large number of cross-chain services and the instability of individual cross-chain transactions, we categorize cross-chain transactions into groups. Each group comprises 100 consecutive cross-chain transaction requests, and the average execution time is calculated to represent the execution time for that group.	
Send rate	The sending rate of cross-chain requests, representing the concurrency and load of cross-chain transactions.	

Experiment 1. We conducted performance testing on the complete cross-chain process (from user request initiation to receiving cross-chain data) of the proposed cross-chain mechanism in this article using JMeter. To evaluate the performance of our solution, we compared the time of the complete cross-chain process with the solutions proposed by Lu et al. (2023) and Xiong et al. (2022). Furthermore, in order to better evaluate whether the proposed solution is suitable for a multi-transaction, high-concurrency ITS cross-chain network, we used the NGSIM dataset to set up 10,000 cross-chain traffic services. The number of concurrent services was set to 50, 100, and 200, respectively, and we conducted tests and statistics on the time of the complete cross-chain process. We divided the 10,000 transactions into rounds of 100 transactions each and calculated their averages for evaluation.

Based on the results shown in Figs. 9 and 10, our proposed solution demonstrated an overall cross-chain average time below 2,500 ms for concurrent transaction numbers of 50, 100, and 200, ranging from 1,868 to 2,409 ms. In comparison, the scheme presented by Lu et al. (2023) showed a stable inter-chain communication delay at around 3,000 ms, and the solution proposed by Xiong et al. (2022) remained stable at around 2,500 ms. In addition, compared with Wu (2021), due to the participation of the relay chain, the burden of inter-chain interaction is increased, and the average cross-chain time overhead of this article is about 10.47 s. In summary, our scheme is more effective than Lu et al. (2023) and Xiong et al. (2022) in terms of cross-chain performance. Furthermore, our scheme exhibits minimal fluctuations between 10–90 transaction rounds, demonstrating excellent stability and suitability for ITS cross-domain network environments.

Figure 9 Cross-chain execution time.

Figure 10 Cross-chain performance comparison.

Experiment 2. To study the efficiency of various algorithms in the cross-chain contract under different concurrent traffic service scenarios, we conducted experiments using Hyperledger Caliper. We simulate different transaction concurrency and loads by controlling the transaction sending rate (Send Rate). We set the transaction sending rate per second in the ranges (100, 650) and (1,600, 2,200), with an interval of 50 transactions per second, to test the impact of different concurrent transaction requests on the performance of the cross-chain contract. We measured the performance of different cross-chain functions, including throughput, average transaction execution time, and CPU resource consumption ratio, as shown in Figs. 11–13. The test results showed that with the increase in cross-chain transaction concurrency, the throughput of the cross-chain algorithms significantly improved. The throughput of the cross-chain read operation reached 2100 TPS, and the average execution time remained stable below 10 ms. The average execution time of the cross-chain write operation was approximately 500 ms, meeting the requirements of ITS for cross-chain read operations.

Figure 11 Cross-chain contract throughput.

Figure 12 Cross-chain contract average execution time.

Figure 13 Cross-chain contract CPU resource consumption percentage.

Experiment 3. We deployed the cross-chain contract on the Ethereum network to test and evaluate the cost of cross-chain transactions. Gas consumption is used to measure the transaction cost, which depends on factors such as the complexity of the smart contract, the computational resources required for execution, and the network transmission cost.

In Table 4, we list the costs of the main functional functions of the cross-chain contract, including gas consumption, corresponding ETH, and USD prices. The table is calculated based on the gas price (20 Gwei) and 1 ETH price ($1,849.84) on July 24, 2023. From the table, we can see that the deployment of the cross-chain contract incurs the highest gas cost, reaching 7,877,145 gas, but this situation only occurs once. In addition, the gas cost for initiating cross-chain transactions and submitting acknowledgment transactions is also high, as these two functions involve parsing cross-chain data, transaction creation, data verification, and complex operations related to the ledger. For sending and creating transactions, the gas spent is directly related to the size of the data contained in the transaction. For transactions such as initialization, setting relay node addresses, and obtaining cross-chain transaction objects, the gas spent is fixed, because these algorithms only need to set addresses or cross-chain transaction ID as parameters.

Table 4 The gas cost of each operation in cross-chain contract and the corresponding price.

Function name	Transaction cost (gas)	ETH	Cost (USD)	
CrossChainContract	7,877,145	0.1575429	291.429158136	
Initialize	113,276	0.00226552	4.1908495168	
startCrossChainTx	550,212	0.01100424	20.3560833216	
getCrossChainTx	65,985	0.0013197	2.441233848	
createCrossChainTxNo	87,937	0.00175874	3.2533876016	
createCrossChainTx	38,746	0.00077492	1.4334780128	
executeCrossChainTx	48,965	0.0009793	1.811548312	
sendAckedTx	687,719	0.01375438	25.4434022992	
setRelayNode	47,360	0.0009472	1.752168448	

Discussion

As shown in Table 5, we conducted a comparison based on five aspects: support for identity authentication, support for complex network scenarios, support for heterogeneous cross-chain, security, and efficiency. Through the comparison, we found that the MyBlockEHR (Sonkamble et al., 2021) and Practical AgentChain (Hei et al., 2022) solutions only implemented homogenous cross-chain on the Ethereum platform, and did not support heterogeneous cross-chain operations. Additionally, these two solutions exhibited execution times exceeding 10 s, indicating relatively low efficiency, and they did not address the issue of cross-chain identity authentication. Although the IBE-BCIOT (Shao et al., 2021) scheme considers both identity verification and data security, it does not support complex network scenarios. The FHTI (Li et al., 2023) solution only supports dual-layer cross-chain operations within the same platform’s consortium chains. In contrast, our proposed solution was tested on the Hyperledger Fabric, FISCO BCOS, and Ethereum platforms, demonstrating that it can meet the requirements for heterogeneous chain cross-chain interaction in intelligent transportation scenarios. It effectively addresses data interoperability across multiple transportation blockchains to some extent, considering aspects like thread parallelism and transaction efficiency. Furthermore, our solution is not limited to the intelligent transportation domain. For example, in the IoT domain, cross-domain data exchange between devices is also a common requirement. By modifying the RNCCP data structure, our solution can be easily extended to this domain, facilitating secure and efficient communication among different IoT devices.

Table 5 Comparison of cross-chain solutions.

Solutions	Support for identity authentication	Support for complex network scenarios	Support for heterogeneous cross-chain	Security	Efficiency	
MyBlockEHR (Sonkamble et al., 2021)	No	Yes	No	Higher	Lower	
IBE-BCIOT (Shao et al., 2021)	Yes	No	Yes	Higher	Higher	
Practical AgentChain (Hei et al., 2022)	No	No	No	Higher	Lower	
FHTI (Li et al., 2023)	Yes	Yes	No	Higher	Medium	
The scheme proposed in this article	Yes	Yes	Yes	Higher	Higher	

Our proposed solution achieved some success in the experiments and demonstrated advantages in cross-chain performance. However, we must also recognize that there are certain limitations and potential challenges. Firstly, although we validated the feasibility of this cross-chain interaction mechanism in our experiments, applying it to real intelligent transportation scenarios may encounter challenges different from the experimental environment. Real-world applications may involve more complex data interaction scenarios and more participants. Ensuring reliability and stability in a real environment requires further research. Secondly, the scalability of the solution is also a concern, especially in real large-scale ITS networks. Maintaining high throughput and performance in such networks will require further investigation. Additionally, the organizations in the ITS cross-chain network may not fully trust each other, and the data in the network may not be entirely open and transparent. This data often contains sensitive information, and there are limitations on the permissions and scope of data sharing between different chains. Therefore, implementing data access control in the ITS network to ensure data and user privacy security is another issue that needs attention.

Conclusion

This article introduces a solution for cross-chain interaction between heterogeneous chains in the ITS and proposes an innovative cross-chain interaction mechanism based on relay nodes. This mechanism is of significant importance in optimizing collaboration and interoperability between heterogeneous chains, enabling secure and efficient data exchange and communication among vehicles, road testing devices, computing nodes, and traffic institutions. By constructing a cross-chain model architecture consisting of relay nodes, relay chains, and cross-domain institutions, as well as designing a secure relay node access and communication scheme based on identity-based encryption, this article provides a feasible solution to the problem of cross-domain data interaction between devices and institutions in ITS. Experimental results demonstrate that this mechanism effectively meets the cross-chain interaction requirements in intelligent transportation scenarios and provides reliable technical support for data sharing and collaboration in ITS. Our solution has been tested and verified on platforms such as Hyperledger Fabric, FISCO BCOS, and Ethereum, confirming its stability and adaptability, thus providing strong support for practical applications.

The cross-chain interoperability mechanism studied in this article facilitates cross-organizational and cross-regional data collaboration in ITS networks, enhancing the value and utility of data. In future work, we will further explore cross-chain data privacy protection schemes. By studying access control mechanisms in cross-chain networks, we aim to ensure the security and privacy of data interactions between devices and traffic institutions across trust domains, while promoting collaboration and information exchange between multiple chains. Additionally, we plan to integrate this mechanism into real-world scenarios (such as intelligent transportation, IoT, etc.) for simulation testing to further validate its effectiveness and reliability in practical applications.

Supplemental Information

Supplemental Information 1 The code of smart contracts.

Click here for additional data file.

Supplemental Information 2 Cross-chain algorithm.

Click here for additional data file.

The authors thank Henan Agricultural University for providing lab facilities used in the implementation of this work.

Additional Information and Declarations

Competing Interests

Author Contributions

Data Availability

The authors declare that they have no competing interests.

Haiping Si conceived and designed the experiments, analyzed the data, authored or reviewed drafts of the article, and approved the final draft.

Weixia Li conceived and designed the experiments, performed the experiments, analyzed the data, performed the computation work, prepared figures and/or tables, authored or reviewed drafts of the article, and approved the final draft.

Qingyi Wang conceived and designed the experiments, prepared figures and/or tables, and approved the final draft.

Haohao Cao conceived and designed the experiments, performed the experiments, authored or reviewed drafts of the article, and approved the final draft.

Fernando Bacao analyzed the data, prepared figures and/or tables, and approved the final draft.

Changxia Sun analyzed the data, authored or reviewed drafts of the article, and approved the final draft.

The following information was supplied regarding data availability:

The code is available at Zenodo.

Weixia Li. (2023). cross-chain. Zenodo. https://doi.org/10.5281/zenodo.8341447.

The data is available at transportation.gov: https://data.transportation.gov/Automobiles/Next-Generation-Simulation-NGSIM-Vehicle-Trajector/8ect-6jqj.

U.S. Department of Transportation Federal Highway Administration. (2016). Next Generation Simulation (NGSIM) Vehicle Trajectories and Supporting Data. [Dataset]. Provided by ITS DataHub through Data.transportation.gov. http://doi.org/10.21949/1504477.

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
