# Peer review of "A secure cross-domain interaction scheme for blockchain-based intelligent transportation systems"

_PeerJ Computer Science, doi:10.7717/peerj-cs.1678_

## Round 0.1 · original submission · Major Revisions

The referral process is now complete. While finding your paper interesting and worthy of publication, the referees and I feel that more work could be done before the paper is published. My decision is therefore to provisionally accept your paper subject to major revisions.

Reviewer 1 ·

Basic reporting

Suggestions for improvement:

Clarify the problem statement: While the paper mentions that secure and efficient data communication is important in the Intelligent Transportation System (ITS), it would be helpful to explicitly state the specific challenges or issues faced in this context. This will provide better context for readers and make the problem statement more precise.

Provide a stronger justification for using blockchain: The paper briefly mentions the benefits of using blockchain for decentralized storage and security, but it would be beneficial to provide a more comprehensive explanation of why blockchain is suitable for addressing the challenges mentioned. This could include discussing the advantages of immutability, transparency, and trust in the context of the ITS.


Discuss the limitations and potential challenges: It would be valuable to address potential limitations and challenges associated with the proposed solution. This could include discussing issues related to scalability, interoperability, performance, or practical implementation considerations. Acknowledging these challenges will provide a more realistic perspective on the proposed solution's capabilities and highlight areas for future research.


Provide a stronger conclusion: The conclusion could be enhanced by summarizing the key findings and emphasizing the significance and potential impact of the proposed mechanism in the context of the ITS. Additionally, discussing potential future research directions or practical implications will add depth and value to the conclusion.

By addressing these areas for improvement, the research paper can enhance its clarity, provide more technical details, and offer a more comprehensive evaluation of the proposed heterogeneous cross-chain interaction mechanism for the Intelligent Transportation System.

Experimental design

Present a more detailed evaluation: While the paper mentions experimental results, it would be beneficial to provide more specific information about the methodology and metrics used for evaluation. Additionally, comparing the results to existing approaches or benchmarks in the field would help demonstrate the superiority or effectiveness of the proposed mechanism.

Validity of the findings

Elaborate on the proposed solution: The paper introduces a heterogeneous cross-chain interaction mechanism based on relay nodes and identity encryption, but it lacks specific details about how these mechanisms are implemented. Providing more information about the technical aspects of the proposed solution, such as the algorithms or protocols used, will enhance the clarity and credibility of the research.

Reviewer 2 ·

Basic reporting

- The paper is well-written and easy to read; no severe errors have been detected with regard to syntax and grammar.

- The structure of the paper is adequate but could be improved. The current structure is Introduction, Background, Proposal, Implementation and Conclusions. It would be nice to have a specific section to analyse state-of-the-art and validation/results to highlight the results obtained from the implementation.

- Literature references are good with enough quality, but a better state of the art must be carried out. That is, certain topics of the background must be further deepened; for instance, libp2p is only described, but no discussion or alternatives are analysed, and smart contracts are only described, but there are a vast number of alternatives of smart contracts and implementations that should be analysed and discussed, please have a look of this references:

Smart Contract Languages: A Multivocal Mapping Study ACM Computing Surveys (CSUR) 54 (1), 1-38
A systematic mapping study on current research topics in smart contracts. Int. J. Comput. Sci. Info. Technol. 9, 5 (Oct. 2017)
Blockchain-based smart contracts: A systematic mapping study. In Proceedings of the CS&IT. Academy & Industry Research Collaboration Center
Smart contract applications within blockchain technology: A systematic mapping study. Telemat. Informat. 35, 8 (2018), 2337–2354.
Use of blockchain smart contracts in software engineering: A systematic mapping. In Proceedings of the ICCSA (5) (LNCS), Vol. 11623.

- The use of figures, tables, and the presentation of data is correct. The environment algorithm style is not correctly used throughout the paper, and Algorithms 1, 2 and 3 are useless.

- The authors use certain pseudo-formalism to present the IBE mechanism and the Relay Node Secure Access and Communication Scheme. The authors should use definitions for the Algorithms G, K, E and D etc. And they have to change the Relay Node Secure Access and Communication Scheme to separate the concepts with regard to definitions to do formalism to the section or eliminate the formalisms entirely.

Experimental design

- The topic and proposal fit perfectly with the aims and scope of the journal.

- Regarding the hypotheses, the authors argue that the current interoperability of transport services (ITS) requires a centralised authentication solution. Therefore, the authors propose a new framework for transport interoperability with mechanisms based on blockchain and identity to improve the transparency, traceability and security in the ITS.

- The proposal is technically well explained and described enough overall, the main concerns, such as the identity and communication scheme, are deeply described con figures, schemes, formalisms, etc. However, the formalisms must be improved, and Algorithms 1,2,3,4 looks more like a piece of code; it would be nice to describe the pseudo-code that enables us to understand the logic behind more than specific instructions of code.

- As previously mentioned, there is a lack of analysis of the state-of-the-art in the form of Related work to compare the solution with others.

- The authors provide results based on a simulation in a controlled environment to check validity and performance. The authors describe the technologies (2 blockchains), conditions, and dataset (NGSIM). However, the authors must clarify more details about the environment and dataset, for instance, the number of nodes used to simulate and the number of transport services simulated, information about the features of the dataset must be given, and it is crucial to provide resources (software, smart contracts and dataset) to enable the replication of the experiments. Further, the authors should discuss the results with regard to the technologies used and the implications of using other blockchain technologies such as Cardano or Ethereum. Regarding the evaluation test, the study is based on performance (time). Still, it would be nice to discuss the implications of consumption in terms of gas or similar since it is very relevant in other blockchains such as Ethereum.

Validity of the findings

- The proposal looks sound relevant, and technologically feasible, as demonstrated. Still, with the results shown, the authors cannot generalise the solution to use in any environment since it is only tested in a simulated scenario.

- The authors must provide resource datasets, smart contracts, and other software to facilitate replicability.

- In my opinion, the authors should undertake several amendments to improve the results in terms of the following:
* Include a comparative with other solutions (Related work)
* Provide resources for the evaluation
* Improve evaluation by taking into account other blockchains (using real test networks) and analysing the consumption in terms of gas.
* Include a Threat of validity section in which authors discuss the limitations and potentials of the solution.

Additional comments

Minor issues:
- Revise the use of open quotes, for instance, ”cross-chain read”, ”cross-chain write”, ”creat

---

## Round 0.2 · Major Revisions

According to the reviewers' comments, there are several issues to be discussed. Thus, my decision is to revise and resubmit the paper.

**Language Note:** The review process has identified that the English language must be improved. PeerJ can provide language editing services - please contact us at copyediting@peerj.com for pricing (be sure to provide your manuscript number and title). Alternatively, you should make your own arrangements to improve the language quality and provide details in your response letter. – PeerJ Staff

Reviewer 1 ·

Basic reporting

Authors updated the paper as per my comments.

Experimental design

As above

Validity of the findings

As above

Reviewer 3 ·

Basic reporting

In this manuscript, a heterogeneous cross-chain interaction mechanism based on relay nodes and identity encryption to solve the problem of data cross-domain interaction between devices and agencies in the ITS was proposed. However, I have the following major concerns.:

There are several writing issues on the paper based on the reference format. It should be the same format. Therefore, all the issues should be checked.

There are a lot of typos in the paper. Authors should review the paper carefully.


Authors give relevants studies however, they should compare the related works to identify and highlight the novelty of this work.

Experimental design

Relay nodes may make a particularly large improvement in the energy efficiency of a wireless network, generally. So, if a protection and encryption method may be improved, can we say that all the system is reliable? May be authors have some good reasoning behind it.


In addition, authors says the task of relay nodes "3) message broadcasting and subscription". If we consider a broadcasting, shouldn't we ensure that all paths from source to destination are reliable? In Algorithm 3.

Validity of the findings

Authors gives avg exec.time performance of the appraoches. If there is an improvement on this point, complexity analysis of the algorithms should be given.

Additional comments

It is just a suggestions to be more clear; authors give related parameters before each of experiment, for example experiment 1, it can be more clear if a parameter or requirements table is/are given.for dataset, services definition.

Reviewer 4 ·

Basic reporting

The authors propose a heterogeneous cross-chain interaction mechanism based on identity encryption as a solution to the interaction problem between data fields in intelligent transportation system.

The authors provide the experimental study and discussion sections in detail and address all concerns. This is acceptable as it is.

Experimental design

The authors provide the experimental study and discussion sections in detail.

Validity of the findings

The present experimental design validates the findings of the proposed method.

---

## Round 0.3 · Major Revisions

One of the reviewers pointed out that he/she was not satisfied with the revision. Please check the revision and give the required responses to the reviewer.

Reviewer 1 ·

Basic reporting

Authors updated the paper as per my comments.

Experimental design

As above

Validity of the findings

As above

Additional comments

As above

Reviewer 2 ·

Basic reporting

Sorry, I want to ensure that it is not my fault, but none of my recommendations have been taken into account, no clues are given in the rebuttals, and no changes are made in this regard within the paper in that direction. Please reconsider my recommendations again.

Experimental design

.

Validity of the findings

.

Reviewer 3 ·

Basic reporting

Is well established and well improved paper in terms of comments.

Experimental design

Authors have done and considered reviewers comments and suggesitons.

Validity of the findings

Authors give complexity analysis and validity explantations.

Additional comments

The paper is generally well-written. The authors use clear language throughout the paper. It includes several references to existing research, which helps readers understand the problem domain and its significance.

Reviewer 4 ·

Basic reporting

The authors propose a heterogeneous cross-chain interaction mechanism based on identity encryption as a solution to the interaction problem between data fields in intelligent transportation system.

The authors provide the experimental study and discussion sections in detail and address all concerns. This is acceptable as it is.

Experimental design

The authors provide the experimental study and discussion sections in detail.

Validity of the findings

The present experimental design validates the findings of the proposed method.

Additional comments

-

---

## Round 0.4 · accepted · Accept

We are happy to inform you that your manuscript has been accepted for publication since the reviewers' comments have been properly addressed.

Reviewer 2 ·

Basic reporting

.

Experimental design

.

Validity of the findings

.

Additional comments

From my side, sorry for the confusion and thanks to the authors for their patience in the process. After revising the changes made, I have to say that the authors have considered all my comments and the results are very good and the quality of the paper has increased a lot. I have to congratulate the authors on the results. In my opinion, the paper is ready to be published now.

Reviewer 3 ·

Basic reporting

The paper has been improved according to the reviews received in the previous rounds and is now fluent and interesting.

Experimental design

The paper has clear and understandable experiment accoring to the previous version.

Validity of the findings

It presents well-written and explained findings.